# T³-S2S: Training-free Triplet Tuning for Sketch to Scene Generation

## Abstract

Scene generation is crucial to many computer graphics applications. Recent advances in generative AI have streamlined sketch-to-image workflows, easing the workload for artists and designers in creating scene concept art. However, these methods often struggle for complex scenes with multiple detailed objects, sometimes missing small or uncommon instances. In this paper, we propose a **T**raining-free **T**riplet **T**uning for **S**ketch-to-**S**cene (**T³-S2S**) generation after reviewing the entire cross-attention mechanism. This scheme revitalizes the existing Control-Net model, enabling effective handling of multi-instance generations, involving prompt balance, characteristics prominence, and dense tuning. Specifically, this approach enhances keyword representation via the prompt balance module, reducing the risk of missing critical instances. It also includes a characteristics prominence module that highlights TopK indices in each channel, ensuring essential features are better represented based on token sketches. Additionally, it employs dense tuning to refine contour details in the attention map, compensating for instance-related regions. Experiments validate that our triplet tuning approach substantially improves the performance of existing sketch-to-image models. It consistently generates detailed, multi-instance 2D images, closely adhering to the input prompts and enhancing visual quality in complex multi-instance scenes.

## 1 Introduction

Scene generation plays a significant role in visual content creation across various domains, including video gaming, animation, filmmaking, and virtual/augmented reality. Traditional methods heavily rely on manual efforts, which require designers to transform initial sketches into detailed multi-instance scene concept art through numerous iterations. Recently, technological innovations such as Stable Diffusion (Rombach et al., 2022; Podell et al., 2023) equipped with ControlNet (Zhang et al., 2023) and integrated with advanced text-to-image technologies (Kim et al., 2023), have streamlined this process. These advancements have notably decreased the workload for designers by automating the conversion of simple sketches into complex scenes. While these technologies perform well with common scenes involving typical instances, they struggle with generating complex multi-instance scenes, particularly with unusual and small instances.

Alternatively, multi-instance synthesis involves incorporating layouts of multiple instances as additional input through bounding boxes, and can effectively manage the generation of multiple instances. However, most methods (Yang et al., 2023; Li et al., 2023; Liu et al., 2023; Sun et al., 2024; Wang et al., 2024b; Zhou et al., 2024) are training-based and require further training when integrated with sketches that contain minimal semantic information, necessitating the collection of numerous scene images. In sectors such as gaming, animation, and film, copyright restrictions significantly hinder scene generation and cannot be disregarded. Conversely, some training-free efforts (Xie et al., 2023; Chen et al., 2024; Feng et al., 2022), like Dense Diffusion (Kim et al., 2023), focus primarily on the impact of the attention map, but they overlook the interaction between the attention map and the value matrices, failing to accurately align with the designer's sketches.

Our strategy involves maintaining ControlNet's sketch-following capabilities while exploring the challenges of synthesizing multiple instances. We aim to develop a training-free tuning mechanism that harnesses the inherent creative capabilities of existing models, eliminating the need for extensive data collection or additional training. We conduct a comprehensive analysis of the cross-attention

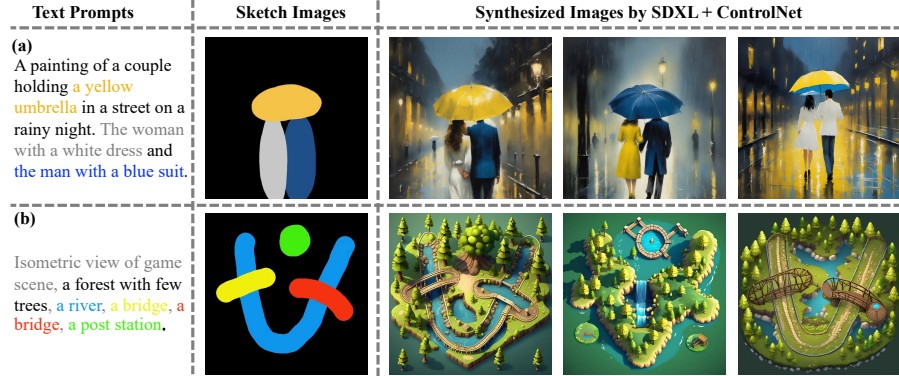

Figure 1: The SDXL-base model (Podell et al., 2023) and ControlNet model (Xinsir, 2023) perform well with common instances like humans, but they struggle with complex multi-instance scenes involving small instances and fail to accurately follow users' prompt.

mechanism, identifying two more issues contributing to the performance gaps in generating detailed scenes beyond the attention maps: imbalanced prompt energy and value homogeneity across the cross-attention layers. These two factors often lead to low competitiveness of unusual instances and high coupling among similar instances, resulting in a final image that deviates from the intended instance prompts.

In this paper, we introduce a **T**raining-free **T**riplet **T**uning for **S**ketch-to-**S**cene (**T**$^3$-**S2S**) generation. Initially, prompt balance improves token representation by adjusting the energy of instance-specific keywords in global text prompts, ensuring that rare instances are adequately represented and remain competitive in the attention mechanism. Subsequently, characteristics prominence distinguishes instance-specific attributes by using a TopK selection strategy from value matrices to amplify feature maps in corresponding channels, highlighting unique instance traits in the multi-channel feature space without extra parameters. Lastly, dense tuning adapted from Kim et al. (2023) is applied in the ControlNet branch to refine the contour information of the attention map to compensate for its suboptimal overall strength of instance-related regions. Together, these three tuning strategies form a cohesive triplet strategy that enhances the entire cross-attention mechanism, balancing token competition, enriching the expression of attention maps, and accentuating each instance's characteristics. Experimental evaluations indicate that our T$^3$-S2S approach boosts the performance of existing text-to-image models, consistently producing detailed, multi-instance scenes that closely align with the input sketches and input prompts.

The key contributions of our work are summarized as follows.

- We investigate the underlying mechanisms of the cross-attention layer and identify the imbalance of prompt energy and homogeneity of value matrices.
- Our T$^3$-S2S model advances a stable diffusion approach by balancing token competition, enriching the expression of attention maps, and accentuating each instance's characteristics.
- Combined with the triplet tuning, our T$^3$-S2S model enhances the representation of unusual and small instances and realizes high-quality generations of complex multi-instance scenes.

## 2 RELATED WORK

**Text-to-Image Synthesis**. In the rapidly evolving field of text-based image generation, various model architectures and learning paradigms have emerged, as highlighted by several key studies (Mansimov et al., 2015; Reed et al., 2016; Xu et al., 2018; Qiao et al., 2019; Zhu et al., 2019; Ramesh et al., 2021; Ding et al., 2021; 2022; Yang et al., 2022; Croitoru et al., 2023; Dhariwal & Nichol, 2021; Kingma et al., 2021). Recently, diffusion models (Rombach et al., 2022; Podell et al., 2023; Saharia et al., 2022) have marked a major breakthrough, significantly improving the fidelity and realism in text-to-image generation, which rely on structured denoising (Ho et al., 2020) with latent diffusion (Rombach et al., 2022). Among these, the SDXL model and its variants, which are widely adopted in both academia and industry, are chosen as the baseline for our work.

**Sketch-to-image Synthesis**. While text-to-image models can generate high-fidelity, realistic images, they struggle to accurately convey complex layouts with text prompts alone. In tasks such as scene design for games, animation, film, or virtual reality, hand-drawn sketches with semantic information provide a more effective way to express design ideas. In the field of diffusion-based generation, notable works include ControlNet (Zhang et al., 2023), Make-a-scene (Gafni et al., 2022), and T2I Adapter (Mou et al., 2023) handle various additional visual conditions, including sketches, while methods like Dense Diffusion (Kim et al., 2023), SpaText (Avrahami et al., 2023) and MultiDiffusion (Bar-Tal et al., 2023) focus specifically on sketch-based inputs. In particular, Dense Diffusion is a training-free approach that adjusts the attention map by amplifying sketch-relevant tokens and downplaying less important ones, allowing the model to better distinguish between instances. ControlNet is a powerful solution for sketch-to-scene generation, recognized for its exceptional ability to accurately follow conditions. However, these models often struggle with complex multi-instance scene generations, particularly when handling unusual or unique instances, and frequently overlook smaller instances. Recently, Xu et al. (2024) proposed an efficient pipeline for automatically generating interactive 3D game scenes from users' natural input sketches using the SDXL and ControlNet models. However, the approach is also limited by the diversity and multi-instance representation in the intermediate 2D isometric image generation.

**Multi-instance Synthesis**. Multi-instance synthesis is closely related to sketch-to-scene generation due to its controllable layout. Training-free modulations (Xie et al., 2023; Chen et al., 2024; Lian et al., 2023; Feng et al., 2022) and training-based fine-tuning methods (Yang et al., 2023; Li et al., 2023; Liu et al., 2023; Sun et al., 2024; Wang et al., 2024b; Zhou et al., 2024) tackle the challenge of diffusion models accurately representing multiple instances with bounding boxes. For example, Li et al. (2023) (GLIGEN) used bounding box coordinates as grounding tokens, integrating them into a gated self-attention mechanism to improve positioning accuracy, while Liu et al. (2023) employed a latent object detection model to separate objects, masking conflicting prompts and enhancing relevant ones. Despite existing methods of generating images with correct positions, these box-based approaches struggle with simple sketch inputs and fail to strictly follow the designer's sketch. Our work leverages ControlNet's sketch-following capabilities and investigates the challenges of synthesizing multiple instances. We aim to design a training-free tuning mechanism to enhance modeling within cross-attention operations, addressing these challenges effectively.

# 3 ANALYSES OF LATENT DIFFUSION

## 3.1 CROSS-ATTENTION MECHANISM

In text-to-image generation tasks, diffusion models aim to transform textual prompts into corresponding images accurately by integrating textual information through cross-attention layers within the UNet model (Rombach et al., 2022; Podell et al., 2023). The mechanism of cross-attention computes attention maps that align intermediate image features with textual embeddings, which can be mathematically represented as:

$$\mathbf{F}_m = \mathbf{A}_m \mathbf{V}_m = \text{softmax}\left(\frac{\mathbf{Q}_m \mathbf{K}_m{}^T}{\sqrt{d_m}}\right)\mathbf{V}_m, \tag{1}$$

where $\mathbf{F}_m$ is the output of the $m$th cross-attention layer, $\mathbf{A}_m$ is the attention map. The query matrices $\mathbf{Q}_m \in \mathbb{R}^{b_m \times d_m}$ are derived from the $m-1$th intermediate representations within the UNet, where $b_m$ represents the spatial dimensions (height multiplied by width) and $d_m$ is the embedding dimension. The key $\mathbf{K}_m \in \mathbb{R}^{n \times d_m}$ and value $\mathbf{V}_m \in \mathbb{R}^{n \times d_m}$ matrices are generated from the encoded text embeddings $\mathbf{S} \in \mathbb{R}^{n \times d}$ from the prompts, where $n$ is the number of tokens and $d$ is the dimension. Based on the mechanism, many explorations (Hertz et al., 2022; Voynov et al., 2023; Feng et al., 2022; Chen et al., 2024) and modulations (Kim et al., 2023; Wang et al., 2024a; Ma et al., 2024; Sun et al., 2024) on attention maps tried to figure out how their behaviors at different layers affect the final generations and utilize training-free modulations as well as fine-tuning strategies to improve generation quality. However, despite the significant focus on attention map optimization, there has been relatively little investigation into the entire process of the cross-attention mechanism. Specifically, the role of the $\mathbf{K}_m$ matrices in shaping the expression of attention maps, and how $\mathbf{V}_m$ contributes to the feature output through the interaction with these maps, remain under-explored.

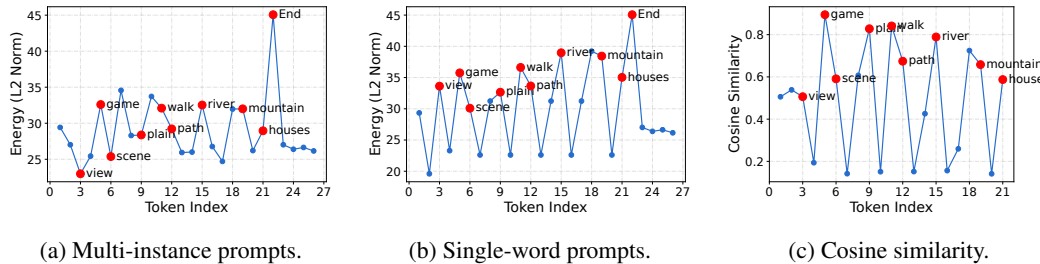

(a) Multi-instance prompts.  (b) Single-word prompts.  (c) Cosine similarity.

Figure 2: Comparison of text embeddings between the prompts ("Isometric view of game scene, a plain, walk path, a river, a high mountain, houses.") and single-word prompts (separate each individual word from the global prompts).

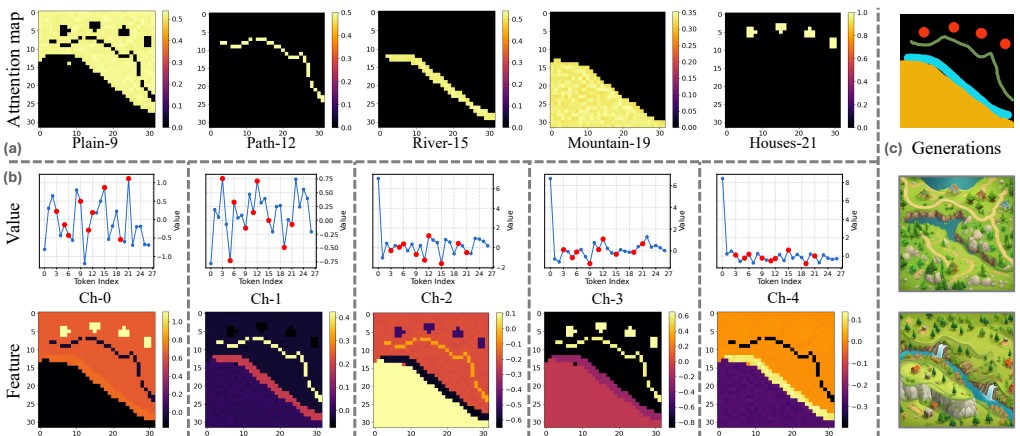

Figure 3: Interaction between attention maps and value matrices with prompts from Figure 2 using dense control (Kim et al., 2023) in the SDXL-base model and the ControNet model. (a) Sketch-relevant attention map generated by Dense Diffusion. (b) Five-channel value-feature pairs. (c) Two synthetic images were generated with the sketches.

## 3.2 IMBALANCE OF PROMPT ENERGY

In extensive practice, two commonly observed phenomena are worth noting: (1) When generating a single instance, the model responds well and rarely misses instances, but in multi-instance generation, some instances are easily lost; (2) In multi-instance generation, if an instance is overlooked, techniques like increasing the prompt weight, such as "(houses:1.5)" in WebUI, can enhance the weight of that prompt after embedding. To quantify this difference, we analyze the text embeddings of multi-instance prompts in Figure 1 (b) and their corresponding single words, using energy (L2 norm) and cosine similarity metrics to measure discrepancies, as shown in Figure 2. For example, in Figure 2a, the word "houses" exhibits lower energy compared to other words, which may explain why the instance "houses" is easily overlooked in Figure 1 (b). Low energy likely results in lower values in the $\mathbf{K}_m$ and $\mathbf{V}_m$ matrices during the transformation from text embeddings, leading to diminished attention. In contrast, words encoded separately in Figure 2b tend to show higher energy levels, which aligns with the observation that single instances are generally well-represented. Additionally, cosine similarity analysis reveals that embedding full sentences alters the distribution of word importance, reducing the emphasis in some instances. Scaling up the embeddings directly enhances the energy levels, thereby increasing the competitiveness of instances in the generation process by boosting their influence in both the attention map $\mathbf{A}_m$ and the value matrix $\mathbf{V}_m$. Understanding the **imbalance of prompt energy** in text embeddings highlights the importance of balancing and scaling energy levels, which offers an interesting perspective to improve multi-instance scene generation.

### 3.3 HOMOGENEITY OF VALUE MATRICES

As a key component of cross-attention, the interaction between attention maps and value matrices determines the characteristics of each feature channel related to multiple instances, such as geometry and attributes. However, this process remains poorly understood due to the inherent noise in both attention maps and generated features. Thereby, inspired by Dense Diffusion (Kim et al., 2023), which enhances sketch-relevant values of the attention map $\mathbf{A}_m \in \mathbb{R}^{b_m \times n}$ depicted in Figure 3 (a), this strategy effectively highlights different instances for each token with a defined level of emphasis. Then, we visualize five-channel value-feature pairs from the $\{\mathbf{v}_m^j \in \mathbb{R}^n\}_{j=1}^{d_m}$ (denoted as $\mathbf{V}_m$) and corresponding feature map $\{\mathbf{f}_m^j \in \mathbb{R}^{b_m}\}_{j=1}^{d_m}$ (denoted as $\mathbf{F}_m$) in Figure 3 (b).

From Figure 3, we can observe: (1) **Extremums matter**: Tokens with extreme values far from zero generate stronger instance characteristics when interacting with attention maps. (2) **Small areas overlooked**: Instances with small areas, such as "Path" and "Houses", are easily neglected in the final image, despite having strong responses in feature maps. This can resemble **homogeneity of values**, where numerical differences between tokens in the value matrix are minimal, and the model struggles to distinguish between instances, leading to instance coupling and the failure to generate certain instances in the final image. This highlights the need for significant numerical disparities among tokens to ensure instance representation.

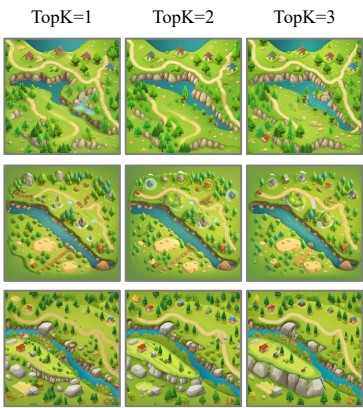

Figure 4: Generations by amplifying the TopK values of the value matrices based on the pipeline in Figure 3.

To assess this potential, we amplify the TopK values in each channel of the value matrices two-fold, as shown in Figure 4. As K increases, particularly at K=2, the model initially generates all instances successfully. However, this also introduces excessive noise, cluttering images with unnecessary details, as seen in the depiction of houses. This experiment suggests that increasing the TopK values enhances token competitiveness and reduces value homogeneity. However, it also highlights a trade-off between instance completeness and visual clarity, underscoring the need for a balanced approach to value amplification in dense diffusion.

## 4 PROPOSED APPROACH

In this section, we present an overview outlining the comprehensive mechanism of our approach. This is followed by a detailed examination of the prompt balance and the characteristics prominence.

### 4.1 OVERVIEW

To address the challenges of the cross-attention mechanism highlighted in Section 3, we introduce the training-free triplet tuning strategy, which builds on the strengths of the pre-trained SDXL and ControlNet models, incorporating textual prompts $\mathbf{c}_g = \{c^i\}_{i=0}^l$ ($l$ is the number of words) and corresponding sketch images $\mathbf{C}_s \in \mathbb{R}^{h \times w}$, as detailed in Figure 5.

The proposed training-free triplet tuning can be divided into the following three modules.
(1) **Prompt Balance**: This module identifies instance keywords within global text prompts, replaces their embeddings with corresponding single-word embeddings, and adjusts the energy of these keyword embeddings to maintain balance. By balancing the energy of the keyword embeddings, the method enhances the representation of instances within key and value matrices. This process improves the competitiveness of instance tokens among all tokens, ensures consistency across instance tokens, and reduces the likelihood of overlooking rare or unusual instances.
(2) **Characteristics Prominence**: This module selects instance-related tokens and their sketches by identifying the TopK values for each channel in the value matrices, creating an instance-specific mask. The mask is then used to scale up the feature map for the corresponding channel. This approach enhances the distinction of instances within the multi-channel feature space without additional parameters, ensuring that instances' characteristics are more prominently emphasized.

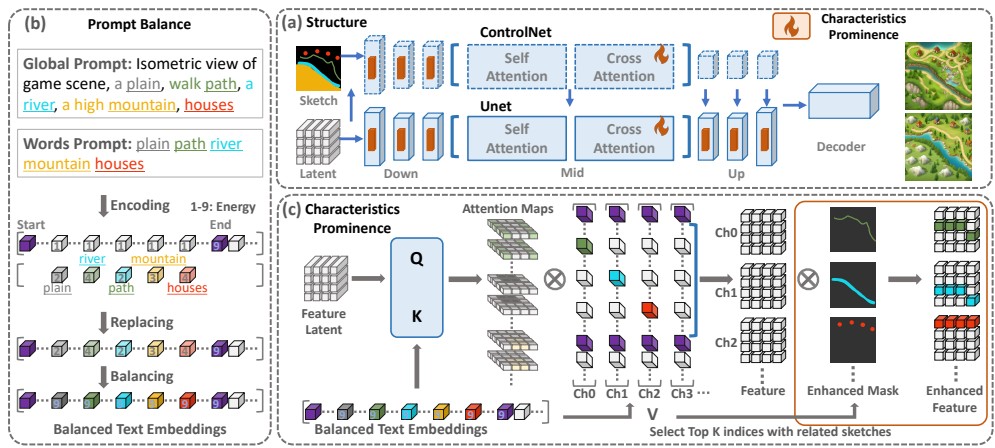

Figure 5: Overview of the proposed training-free triplet tuning strategy in the frozen pre-trained latent diffusion model. (a) The orange parts indicate the proposed module plugged into the ControlNet and U-Net framework. (b) The left part shows the energy tuning of prompt balance. (c) The bottom part indicates the training-free tuning of characteristics prominence.

(3) **Dense Tuning**: While prompt balance increases the strength of the embedding matrices related to instances, enhancing their competitiveness in the attention map, the overall strength of the attention map remains suboptimal. Meanwhile, given that more contour information resides in the ControlNet branch, we employ dense modulation directly within this branch to augment the attention map for better modulation. Specific implementation refers to Dense Diffusion (Kim et al., 2023).

Building on these three modules, a unified training-free triplet tuning strategy is implemented throughout the entire cross-attention mechanism. This ensures that the final generation effectively responds to both text and sketch inputs, thereby enhancing the stability and diversity of the generated outputs. In the subsequent section, we will provide a detailed explanation of two newly designed modules and their underlying rationales.

## 4.2 PROMPT BALANCE

As discussed in Section 3.2, the imbalance of prompt energy related to instances influences representations of key and value matrices, and causes instances missing in the generated image. Therefore, by enhancing the balance of keywords' energy and scaling up the values, we can improve the instance accuracy and details in the generated images. To do this, we propose a plug-in strategy for the text embeddings shown in Figure 5 (b), named **prompt balance**.

Specifically, we use an NLP network (e.g., the SpaCy library) to identify instance keywords from the global text prompts $\mathbf{c}_g = \{c^i\}_{i=0}^l$, resulting in reorganized instance keyword prompts $\{c^i\}^{\mathbf{q}}$, where $\mathbf{q}$ is the **indices vector** of keywords in $\mathbf{c}_g$. Then, we encode both the global text prompts and each instance keyword prompts separately into text embeddings $\mathbf{S}_g = \{\mathbf{s}_g^i\} \in \mathbb{R}^{n \times d}$ and $\{\mathbf{s}_w^i \in \mathbb{R}^{1 \times d}\}^{\mathbf{q}}$ by a text encoding network. Next, we replace the embedding of keywords in $\mathbf{S}_g$ with the single-word embedding of $\mathbf{S}_w$ to form a new combined embedding $\mathbf{S}_r \in \mathbb{R}^{n \times d}$, defined as:

$$\mathbf{S}_r = \{\mathbf{s}_r^i\} = \{\mathbf{s}_w^i\}, \quad \text{if } y \in \mathbf{q}, \quad \text{otherwise } \{\mathbf{s}_g^i\}.$$

Generally, the special "end of text" token (located $i_{\text{end}}$) always has the maximum energy as shown in Section 3.2, which could be the upper bound for us to scale up the embeddings of the keywords in $\mathbf{S}_r$ that all keywords have balanced energy relative to the "end of text" token embedding, mathematically represented as:

$$\{\mathbf{s}_r^i\}^{\mathbf{q}} = \{(E_r^{i_{\text{end}}}/E_r^i) \cdot \mathbf{s}_w^i\}^{\mathbf{q}}, \quad \text{where} \quad E_r^{i_{\text{end}}} = \|\mathbf{s}_r^{i_{\text{end}}}\| \text{ and } E_r^i = \|\mathbf{s}_r^i\|.$$

Finally, the balanced text embeddings, denoted as $\mathbf{S}_b$, will benefit the values of instance-based tokens in the key and value matrices, as well as the attention map. Subsequently, it will enhance the competitiveness of instance tokens among all tokens while maintaining consistency across instance tokens, providing a concise summary that highlights the importance of each instance.

### 4.3 CHARACTERISTICS PROMINENCE

While utilizing balanced text embeddings helps balance the competition among instances, diffusion models still face challenges with entity coupling in cross-attention layers, lacking a mechanism to address the interaction between the attention map and value matrices. As discussed in Section 3.3, increasing the TopK values in each channel of the value matrices reduces value homogeneity, but requires a trade-off between instance completeness and noise clarity. In this subsection, we introduce a characteristics prominence technique, applied after feature map computation in the original cross-attention layers, without introducing any additional trainable parameters.

Specifically, instead of directly enhancing the TopK values along the $n$ dimension in the value matrix $\mathbf{V}_m \in \mathbb{R}^{n \times d_m}$, we apply enhancement based on the indices of the TopK values on the feature map $\mathbf{F}_m \in \mathbb{R}^{b_m \times d_m}$ (before the residual adding). For each channel in the value matrix $\mathbf{V}_m$, we find the indices of the TopK values across all valid tokens (between "start of text" and "end of text" tokens):

$$\mathbf{Y}_K = \{\mathbf{y}_K^j\}_{j=0}^{d_m} = \mathrm{TopK}(\mathrm{abs}(\mathbf{V}_m[1:i_{end}]), K) \in \mathbb{R}^{K \times d_m},$$

where $K$ is the number of top values considered. For $j$th channel $\mathbf{f}_m^j \in \mathbb{R}^{b_m}$ in $\mathbf{F}_m$, we check whether each index $i$ in $\mathbf{y}_K^j$ belongs to the instance keyword vector $\mathbf{q}$. If it does, the index $i$ corresponds to a specific instance token $i \in \mathbf{q}$. Then the sketch $\mathbf{u}_m^i \in \mathbb{R}^{b_m}$ of the instance at the current scale will be summed together to generate an enhancement mask $\mathbf{h}_m^j$ for the $j$th channel:

$$\mathbf{h}_m^j = \sum \mathbf{u}_m^i, \quad \text{if } i \in \{\mathbf{y}_K^j \text{ and } \mathbf{q}\}.$$

The whole mask matrices $\mathbf{H}_m = \{\mathbf{h}_m^j\}_{j=0}^{d_m} \in \mathbb{R}^{b_m \times d_m}$ are used to proportionally scale up the corresponding values in the feature map $\mathbf{F}_m$ by a factor $\beta$, obtaining the enhanced feature map $\hat{\mathbf{F}}_m$:

$$\hat{\mathbf{F}}_m = \mathbf{F}_m + \beta \cdot \mathbf{H}_m \odot \mathbf{F}_m,$$

where $\odot$ denotes element-wise multiplication. This enhancement emphasizes instance tokens within the multi-channel feature space, aiding in distinguishing each instance more effectively. The characteristics prominence technique strengthens the attention mechanism by ensuring that each instance is highlighted, even when its sketch is small. By amplifying relevant regions in the feature map, the model improves instance differentiation, making it better suited for multi-instance scene generation.

## 5 EXPERIMENTS

### 5.1 IMPLEMENTATION DETAILS

**Baselines.** We leverage the sketch-processing capabilities of the *SDXL-base* model (Podell et al., 2023) and the *ControlNet* model (Zhang et al., 2023; Xinsir, 2023), serving as our foundational models. Additionally, we extend our comparison to include two sketch-oriented approaches: the training-based *T2I Adapter* (Mou et al., 2023) and the training-free *Dense Diffusion* (Kim et al., 2023), both integrated with the SDXL-base model.

**Setup.** In the triplet tuning scheme, the prompt balance module is integrated into the text encoding process, while the characteristics prominence modules are incorporated across all cross-attention layers. Additionally, the dense tuning module is specifically added to the "down_blocks 2" layers and the "mid_blocks 0" layers within the ControlNet branch. The TopK value $K$ is set to 2, and $\beta$ is kept at 1. During inference, we use the default Euler Discrete Scheduler (Karras et al., 2022) with 32 steps and a guidance scale of 9 at a resolution of $1024 \times 1024$. All experiments are conducted on a single Nvidia Tesla V100 GPU.

**Metrics.** Given that our current approach involves sketch-based multi-instance scene generation, existing benchmarks should be adjusted for our evaluation, such as adding sketch inputs for T2I-CompBench (Huang et al., 2023). Therefore, we design 20 complex sketch scenes, each with more than four sub-prompts, encompassing various terrains (plains, mountains, deserts, tundra, cities) and diverse instances (rivers, bridges, stones, castles). We utilize CLIP-Score (Hessel et al., 2021) for the global prompt and image, and evaluate the CLIP-Score for each background prompt and instance prompts by cropping the corresponding regions. Additionally, we conduct a user study to assess different variants of our approach, using a 1-5 rating scale to evaluate image quality, placement, and prompt-image consistency. Details can be found in **Appendices** C and E.

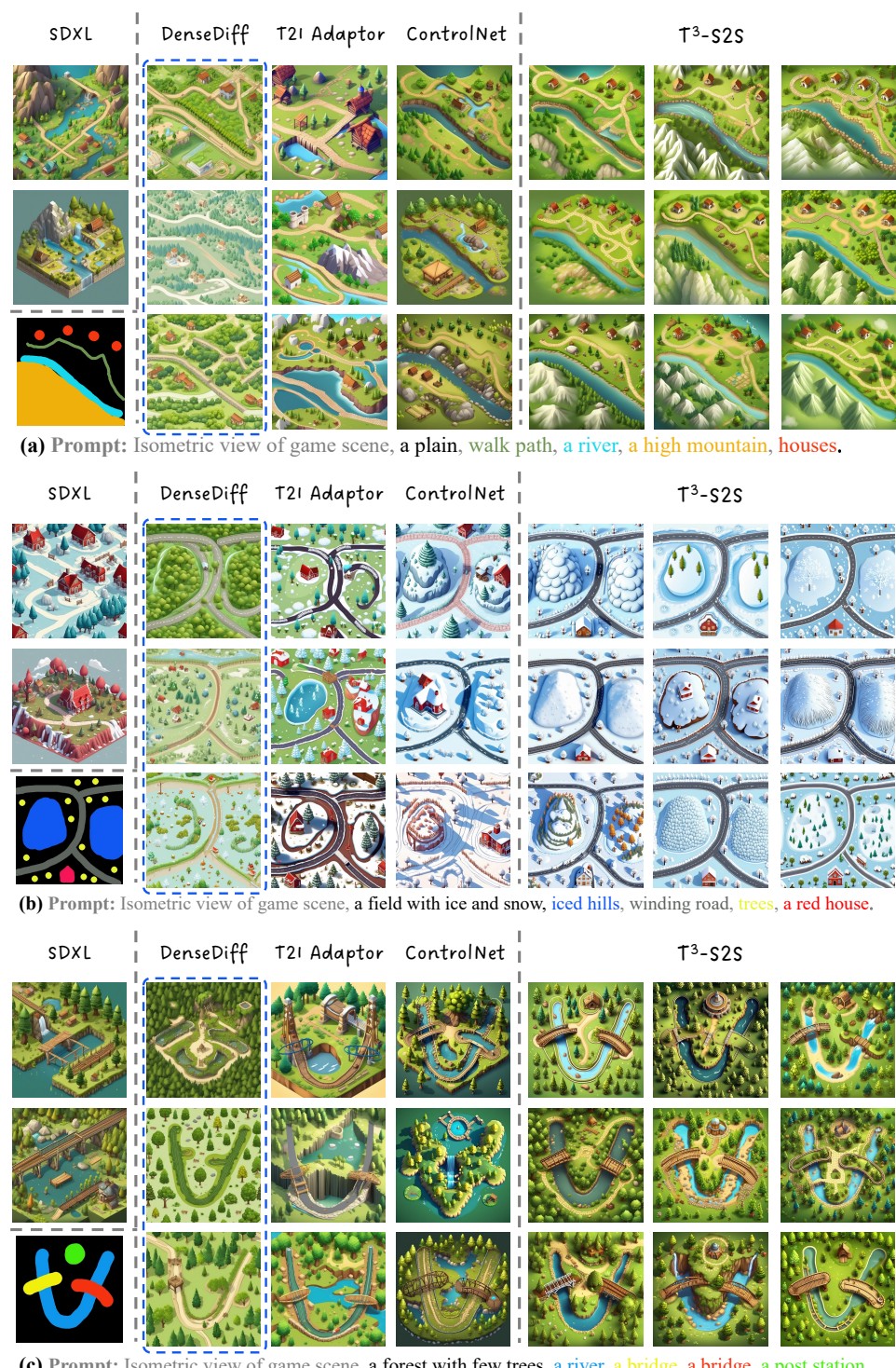

Figure 6: Qualitative comparison with baseline methods. (a) T³-S2S performs well for smaller instances like "houses" and "path", and unusual "mountain". (b) T³-S2S performs well with a large number of small instances "trees". (c) T³-S2S decouples the overlap of instances. Note that the original Dense Diffusion (Kim et al., 2023) based on SD V1.5 (Rombach et al., 2022), has limited prompt response capabilities. For a fair comparison, we apply it to the SDXL model.

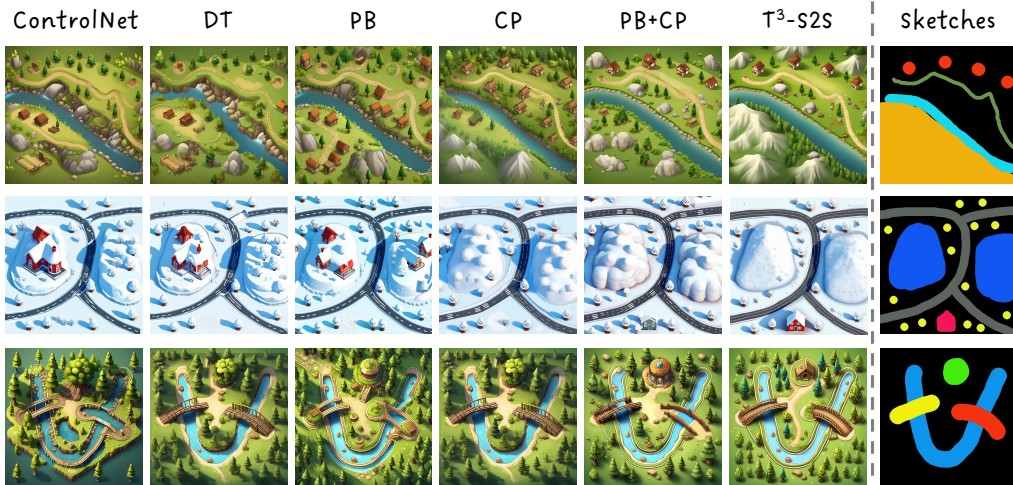

Figure 7: Visual comparison of different inserted modules. DT: Dense Tuning; PB: Prompt Balance; CP: Characteristics Prominence.

## 5.2 MAIN RESULTS

**Qualitative Evaluation.** Building upon the scene design, we develop three representative and complex multi-instance scene scenarios, each incorporating a diverse array of elements to foster varied interactions. We evaluate several approaches, with visual comparisons displayed in Figure 6. Due to the specialized nature of this task, most existing solutions are inadequate, often overlooking small objects and failing to manage instance overlap effectively. When combined with the triplet tuning strategy, our T³-S2S method improves the generation performance of existing SDXL models. For example, Figure 6 (a) showcases the enhanced detail in smaller instances such as "houses" and "path", and even less common elements like "mountains". Similarly, Figure 6 (b) illustrates the effective generation of numerous small instances, like "trees". Furthermore, our approach excels in scenarios with complex instance contour interactions, as depicted in Figure 6 (c), accurately capturing and displaying all details. By leveraging the triplet tuning strategy and an advanced cross-attention mechanism, our approach consistently generates detailed, multi-instance scenes that closely adhere to the original sketches and prompts, ensuring both stability and diversity in generations. Additional game scenes and diverse scenes are provided in **Appendices** C and D.

**Quantitative Evaluation.** We compare CLIP-Scores for global image, instances, and background across different variants and the base ControlNet. A user study is also conducted with a 1-5 rating scale. As shown in Table 1, our approach demonstrates superior performance on the 20 complex multi-instance scenes, with improved fidelity and precision in aligning with text prompts and sketch layouts. The PB module shows modest improvement, while the CP and DT modules provide significant and comparable enhancements. Combining these components allows our T³-S2S approach to achieve a well-balanced outcome.

Table 1: Comparison of CLIP-Score across several variants, evaluated on whole images, masked instance regions, and masked background regions. Includes user study ratings on a scale of 1-5.

| Model | Global↑ | Instances↑ | Background↑ | User↑ |
|---|---|---|---|---|
| ControlNet | 0.3440 | 0.2423 | 0.2539 | 2.34 |
| PB | 0.3447 | 0.2479 | 0.2562 | 2.64 |
| DT | 0.3433 | 0.2463 | 0.2549 | 3.29 |
| CP | 0.3467 | 0.2490 | 0.2563 | 3.45 |
| PB+CP | 0.3465 | 0.2548 | 0.2573 | 3.61 |
| T³-S2S | **0.3497** | **0.2563** | **0.2588** | **3.88** |

## 5.3 ABLATION STUDY

**Module Comparison.** In this validation, we conduct an ablation study to assess the individual and combined impacts of different modules, with the findings detailed in Figure 7. While each module contributes to improving generation quality, no single module fully resolves all challenges: (1)

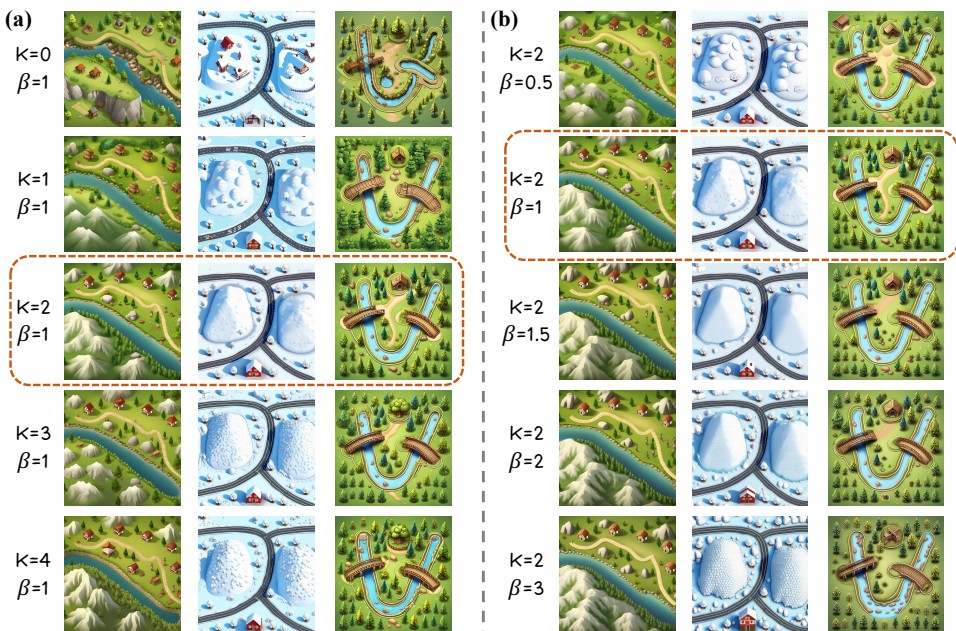

Figure 8: Visual comparison of two hyper-parameters $K$ and $\beta$ suggests that setting $K = 2$ and $\beta = 1$ is a favorable choice.

Dense tuning effectively restricts instance overlap within sketch areas, such as "bridges", by optimizing instance overlap. (2) Prompt balance enhances the visibility of smaller objects like "houses", although it may inadvertently introduce noise associated with these houses. (3) Characteristic prominence sharpens the distinct features of instances, enhancing clarity and reducing irrelevant noise. (4) A combined application of prompt balance and characteristic prominence effectively addresses most issues, approaching success. When these three modules are integrated to form our triplet tuning approach, they enhance the alignment between generated scene images and their corresponding sketches and prompts, leading to more consistent and accurate representations. To further verify the functions of modules, we transfer the PB module to the Attend-and-Excite (Chefer et al., 2023) in **Appendix** F, and the T$^3$-S2S to the T2I adapter (Mou et al., 2023) in **Appendix** G.

**Hyper-parameter Comparison.** To validate our hypothesis, we examine the impacts of varying $K$ and $\beta$ values on generations, with visual results presented in Figure 8. In a set of experiments where $\beta$ is fixed at 1, we find that increasing $K$ initially improves generation quality but eventually leads to more noise. Conversely, when $K$ is held steady at 2, adjusting $\beta$ above 1 consistently produces favorable outcomes, maintaining stable generation quality across higher $\beta$ values. Based on these observations, we determine that the optimal settings for our model are $K = 2$ and $\beta = 1$. We also conduct an analysis of TOP $K$ distribution in **Appendix** B.

## 6 CONCLUSION

In conclusion, our study on the training-free triplet tuning for sketch-to-scene generation has enhanced the ability of text-to-image models to process complex, multi-instance scenes. By incorporating prompt balance, characteristics prominence, and dense tuning, we have effectively addressed issues such as imbalanced prompt energy and value homogeneity, which previously resulted in the inadequate representation of unusual and small instances. Our experimental results confirmed that our approach not only preserves the fidelity of input sketches but also elevates the detail of the generated scenes. This advancement is vital in fields like video gaming, filmmaking, and virtual/augmented reality, where precise and dynamic visual content creation is crucial. Facilitating more efficient and less labor-intensive generation processes, our model offers a promising avenue for future developments in automated sketch-to-scene transformations.

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

## A    DISCUSSION

While our approach is innovative and enhances multi-instance scene generation, it also has some room to improve, primarily stemming from the inherent capabilities of the base model. One significant challenge is the generation of detailed instances, such as textures and finer details. This issue largely arises from the limited understanding of complex descriptions by the CLIP models. Moreover, the characteristic prominence module tends to focus on instance tokens while neglecting some descriptive adjectives. Additionally, our method struggles with accurately capturing very large scenes (exceeding $4096 \times 4096$ pixels) such as expansive game maps, which often include complex relationships like overlaps and interactions between instances. These complex and dynamic scenarios require further enhancements and refinements in our approach to effectively represent and capture such intricate relationships. Building on our current achievements, we plan to further explore these areas in future work to improve detailed multi-instance sketch-to-scene generation.

## B    TOP $K$ ANAYLSIS.

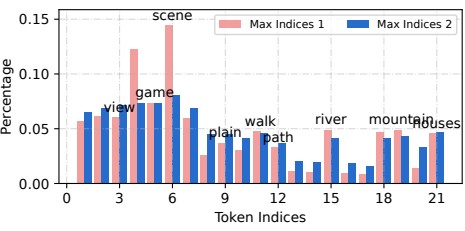 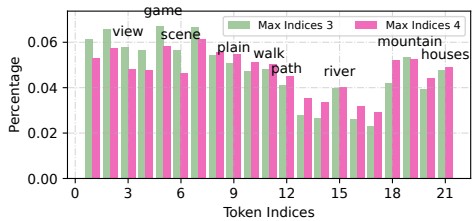

(a) The first and second extreme points.      (b) The third and fourth extreme points.

Figure 9: Histogram of the distribution of indices where the extremum points are located.

In the module of characteristic prominence, two hyperparameters, $K$ and $\beta$, require meticulous tuning. $K$ determines the indices of extreme values within the value matrices. For our analysis, we save these indices and construct a histogram, as depicted in Figure 9. We observe that the probability of later instance tokens achieving the maximum and second maximum values is comparatively lower than that of earlier tokens. Thus, increasing the value of later instance tokens will be beneficial for their representations. However, at the third and fourth extremes, the probabilities tend to converge, indicating that not every token is essential for defining key characteristics. Increasing values for later instances at this point would introduce additional noise. Therefore, setting $K = 2$ is advisable based on the observed trends. For $\beta$, which enhances the characteristics of instances within the feature matrix, an initial increase is beneficial. Nonetheless, there is a critical threshold beyond which further increases in $\beta$ begin to disrupt the distribution within the value matrices.

## C    SKETCH VISUALIZATIONS OF QUANTITATIVE EXPERIMENTS

Using a game scene as an example, we begin each prompt with 'Isometric view of a game scene' to generate controlled synthetic images for game settings. This helps maintain a consistent angle and style, ignoring any incoherent instance sketches that might appear in real-world scenes, thereby focusing on object placement and verifying text-image consistency. We generate all 20 complex scenes using hyperparameters identical to those used in the main results (Figure 6), shown in Figures 12 and 13. The colored sketches are used solely to distinguish between different instances, and the colors used are arbitrary without class or semantic information. To validate this, we also use grayscale sketches as input, and the resulting images are nearly identical under the same random seed (two columns pointed by the red arrows in Figure 12). Meanwhile, our approach is not limited to game scenes. We also test prompts without the fixed game scene phrase, resulting in more diverse angles and styles while maintaining the same quality in object placement and text-image consistency (One row pointed by the green arrows in Figures 12).

## D VISUALIZATION OF DIVERSE SCENES

In the above experiments, we primarily validate the controllability of our method for multi-instance generation in game scenes. However, this does not imply that our approach is limited to game scenarios. To further verify its capabilities, we design three sets of diverse scenes: (1) four common simple scenes; (2) two indoor scenes; and (3) three scenes featuring instances of the same type but with different color attributes. Without changing any hyperparameters, generations are presented in Figure 14. In common scenes, our method effectively mitigates instance overlap under ControlNet control, while in indoor scenes, it handles varied layouts well. For the challenging task of differentiating attributes within identical instances, our approach assigns distinct properties accurately. However, for uncommon attributes like generating a red cat, our method struggles due to limitations inherent in the original SDXL model.

## E METRIC OF USER STUDY

We conduct a user study on 20 scenes, each with 6 variants, generating 100 images per scene. A Gradio-based evaluation interface is designed, which randomly selects one image from 120 sets to create a sub-evaluation system, with images presented anonymously. 23 participants independently rate the images based on the following scale:

- **5**: All instances are accurately placed, and overall image quality is high.
- **4**: One instance is missing or misplaced, or All are placed with lower quality.
- **3**: Two or three instances are missing or misplaced, or placed with lower quality.
- **2**: Three or four instances are missing or misplaced, or placed with lower quality.
- **1**: Multiple instances are missing, with low overall quality.

This detailed rating system helps assess both the accuracy of instance placement and the quality of generated images, whether the generations are aligned with text prompts and sketch layouts.

## F TRANSFER PROMPT BALANCE TO ATTEND-AND-EXCITE

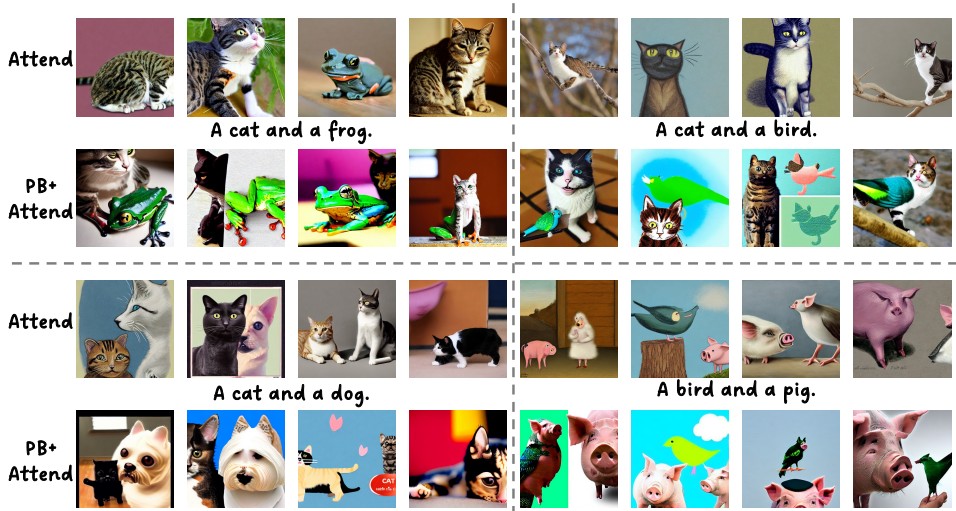

Figure 10: Visualizations for transferring PB to Attend-and-Excite (Chefer et al., 2023). In most cases, both instances are successfully generated. The frog-leg cat and the bird-wing pig further demonstrate the effectiveness since they lack the layouts to separate the instances spatially.

To further validate the PB module, we integrated it into the Attend-and-Excite method (Chefer et al., 2023), based on attention tuning using the SD V1.4 model. The results are shown in Figure 10.

Despite the limitations of SD V1.4, the PB module effectively balances embedding strength between the two instances in scenarios without layout guidance, enhancing their representation. In most cases, both instances are successfully generated. However, in some cases, the attributes of the two objects become entangled, leading to artifacts such as a cat with frog legs or a pig with bird wings, due to the lack of spatial separation, which further demonstrates the effectiveness of the PB module.

# G TRANSFER $T^3$-S2S TO T2I-ADAPTER

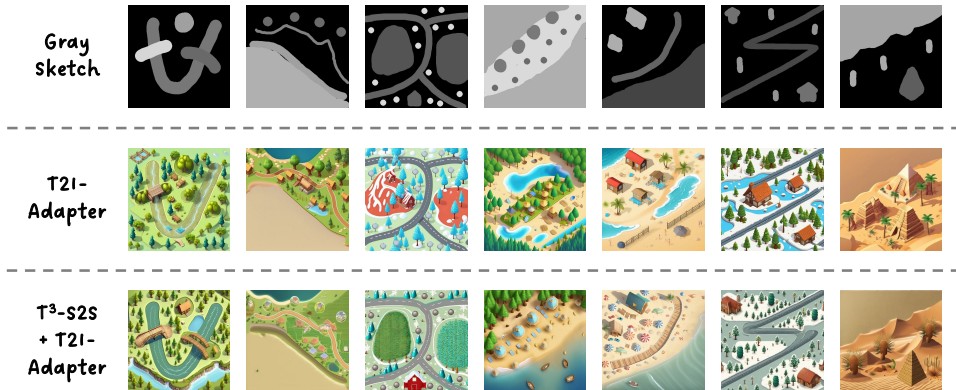

Figure 11: Visualizations for transferring $T^3$-S2S to T2I-Adapter (Mou et al., 2023). $T^3$-S2S effectively improves the T2I-Adapter's alignment with prompts and layouts in complex scenes, demonstrating its control capabilities across different models.

To validate the general applicability of our approach beyond the ControlNet model, we apply $T^3$-S2S to another controllable T2I-Adapter (Mou et al., 2023) model. Although the T2I-Adapter performs best with detailed sketches, we use grayscale sketches for quick validation, which contain less semantic information. We keep the PB and CP modules unchanged, while the DT module is integrated into the SDXL main channel, similar to CP, as it can not be placed in a separate branch like in ControlNet. We use the same prompts and sketches from the main results (Figure 6) and Appendix C, with all other hyperparameters unchanged. The results are shown in Figure 11. $T^3$-S2S effectively improves the T2I-Adapter's alignment with prompts and layouts in complex scenes, demonstrating its control capabilities across different models. However, the generation quality still lags behind the ControlNet-based approach, indicating the need for parameter tuning specific to the T2I-Adapter's distribution and improved sketch inputs to align with the T2I-Adapter. Despite these limitations, the results show that $T^3$-S2S has promising generalizability and can effectively control both ControlNet and T2I-Adapter models.

# H 3D GAME SCENE

Figure 15 demonstrates examples of 3D scenes generated using our method. Building on the approach in (Xu et al., 2024), our method can be used to reconstruct a 3D mesh and further serve as the foundation for generating high-fidelity 3D scenes within the game environment. Similar to (Xu et al., 2024), we also adopt the Depth-Anything-V2 (Yang et al., 2024) method to infer scene depth and reconstruct the complete mesh using the Poisson reconstruction technique.

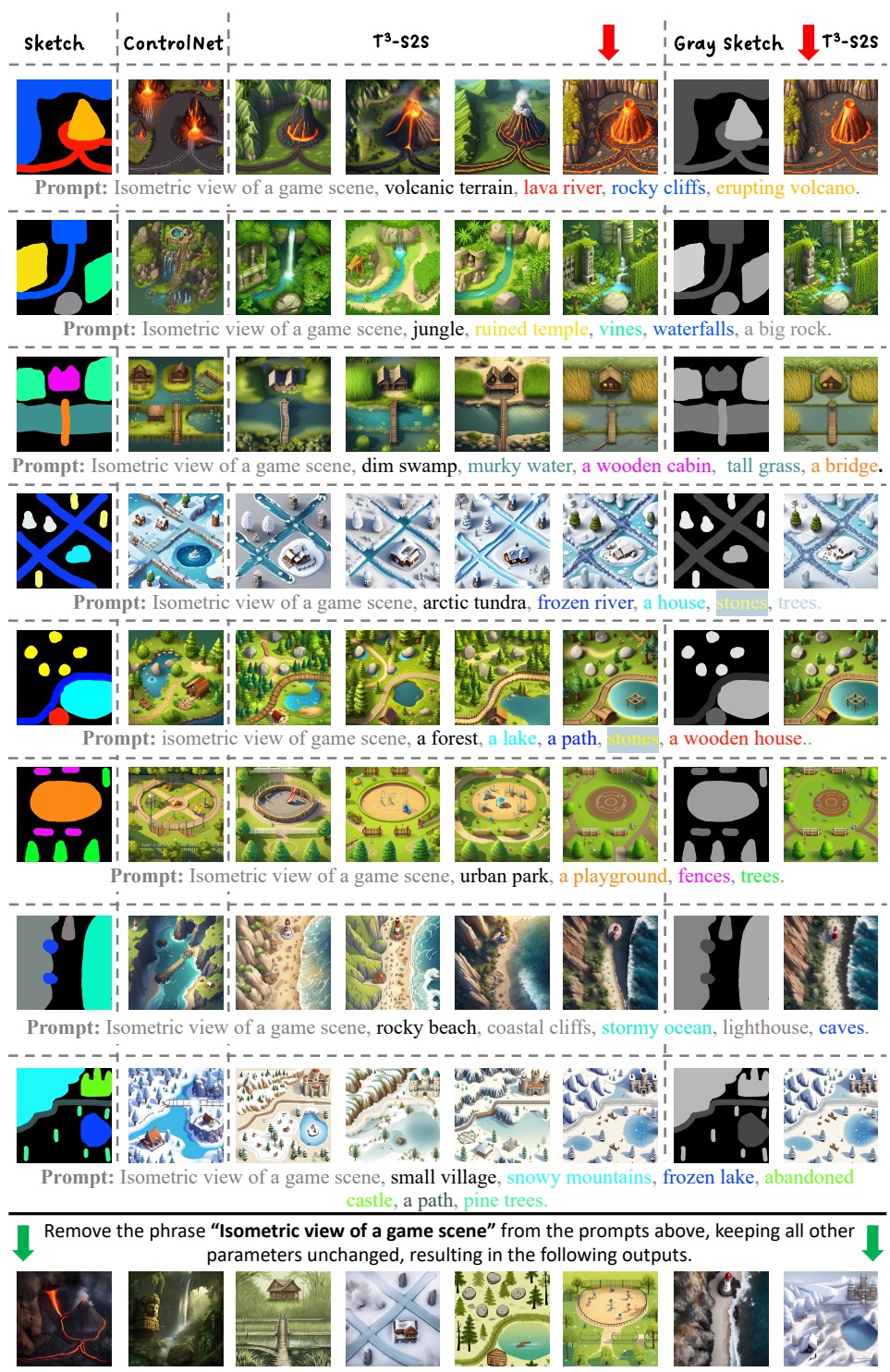

Figure 12: Example results from a subset of the 20 complex scene composition tested using hyper-parameters identical to those used in the main results (Figure 6). (1) Two columns pointed by the red arrows represent the generations using colored and grayscale sketches under the same random seed. (2) One row pointed by the green arrows indicates the generations without the fixed game scene phrase.

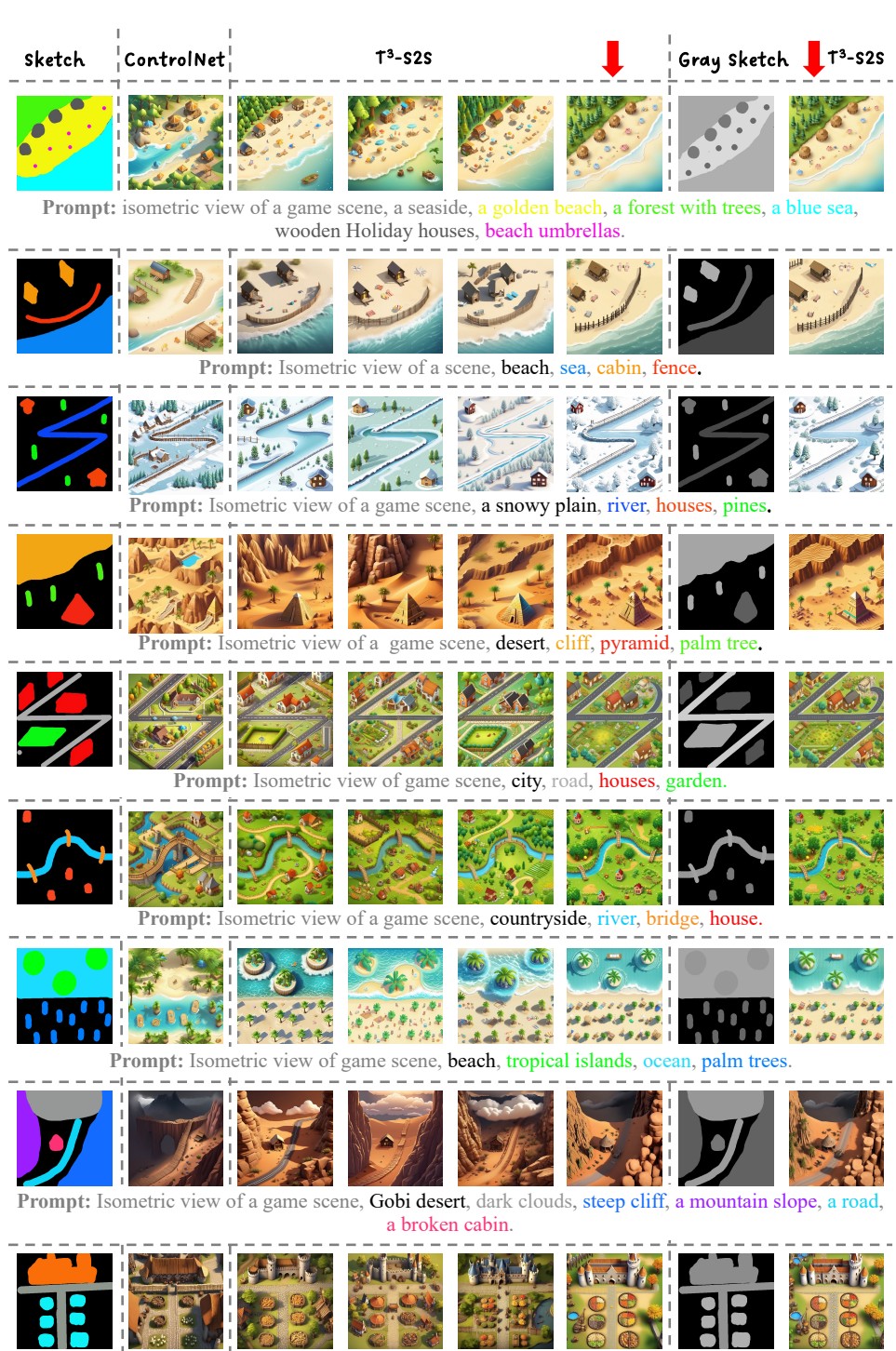

Figure 13: Example results from a subset of the 20 complex scene composition tested using hyper-parameters identical to those used in the main results (Figure 6). Two columns pointed by the red arrows represent the generations using colored and grayscale sketches under the same random seed.

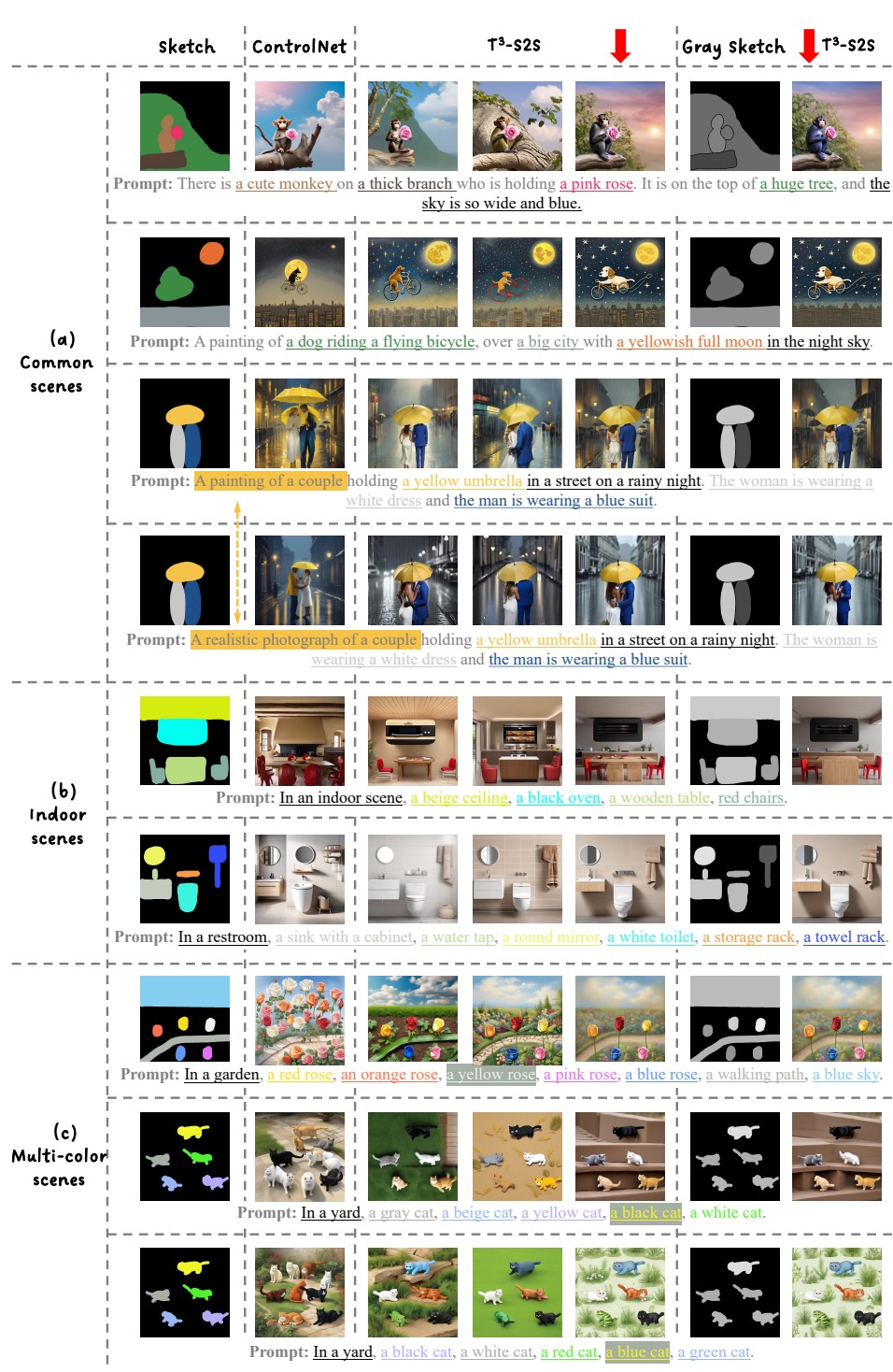

Figure 14: Examples of generated scenes across different settings. (a) Common simple scenes demonstrating effective instance representations under ControlNet control. (b) Indoor scenes showcasing robust handling of varied instance layouts. (c) Scenes with identical instances but different color attributes illustrate precise differentiation of properties.

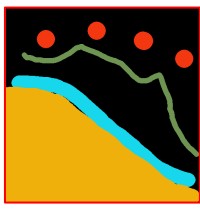 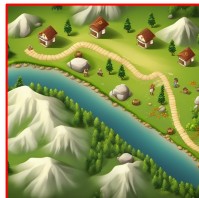 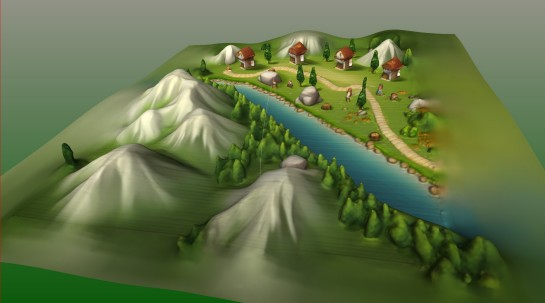

**Prompt:** Isometric view of game scene, a plain, walk path, a river, a high mountain, houses.

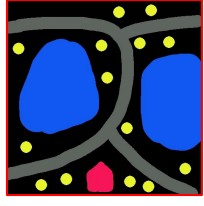 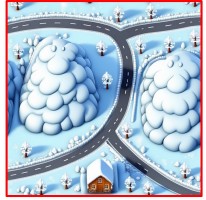 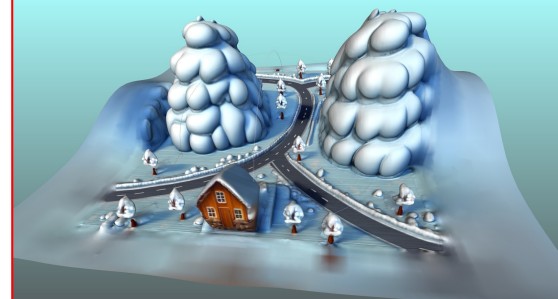

**Prompt:** Isometric view of game scene, a field with ice and snow, iced hills, winding road, trees, a red house.

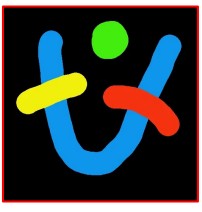 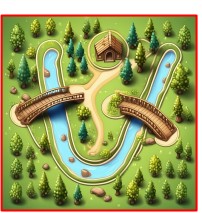 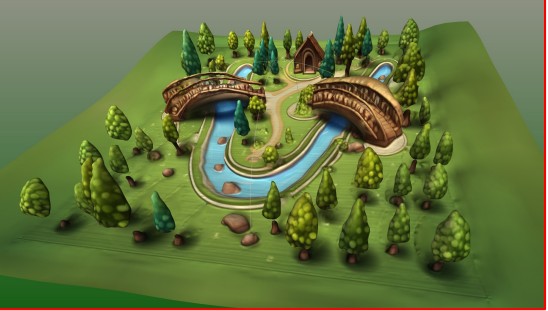

**Prompt:** Isometric view of game scene, a forest with few trees, a river, a bridge, a bridge, a post station.

Figure 15: **Example of 3D scene generation results.** The left side displays the input sketches and text, along with the generated isometric images. The images on the right are rendered from the reconstructed 3D scene using the isometric images.

