# OpenReview forum: "T$^3$-S2S: Training-free Triplet Tuning for Sketch to Scene Generation"
_ICLR.cc/2025/Conference — ICLR 2025 Conference Withdrawn Submission_

### Official Review · Reviewer_wtqN · 2024-10-29

**Soundness:** 2
**Presentation:** 2
**Contribution:** 2
**Rating:** 8
**Confidence:** 5

**Summary:**

This paper proposes an approach, termed Triplet Tuning for Sketch-to-Scene (T3-S2S), a training-free method designed to enhance T2I generation capabilities in multi-instance scenarios, effectively mitigating challenges such as instance omission. The T3-S2S method introduces a prompt balance mechanism to automatically balance the energy of each instance within text embeddings and incorporates a characteristics prominence mechanism that strengthens cross-attention by highlighting Top-K indices within each channel, ensuring that essential features are more robustly represented based on token sketches. Methodologically, T3-S2S demonstrates certain innovations compared to previous approaches, and the results presented by the authors indicate that this method indeed yields notable improvements. Therefore, at this stage, I would rate this paper as [weak accept]. However, I believe the current experimental results are still insufficient; please see the question section for details. I look forward to the authors' responses.

**Strengths:**

Methodologically, T3-S2S demonstrates certain innovations compared to previous approaches, and the results presented by the authors indicate that this method indeed yields notable improvements.

**Weaknesses:**

I believe the current experimental results are still insufficient. First, the paper does not include comparisons with state-of-the-art (SOTA) methods on popular benchmarks. Additionally, it lacks quantitative comparison results, which are essential for a comprehensive evaluation of the method’s effectiveness.

**Questions:**

(1) The prompt balance mechanism could also be applied to text-to-image models. I recommend testing the effectiveness of the prompt balance mechanism on some multi-instance text-to-image benchmarks. Specifically, you could compare it with training-free optimization algorithms designed for multi-instance T2I scenarios, such as AAE [1] and StructureDiff [2].

(2) When performing the characteristics prominence mechanism, the paper states that the optimal setting for K is 2. When the number of instances exceeds K, how does your characteristics prominence mechanism ensure that each instance is effectively enhanced? More specifically, how does it handle multiple instances that belong to the same category but have different fine-grained attributes? For example, when dealing with five cats—one red, one blue, one green, one black, and one white.

(3) I have some questions regarding specific details. When performing characteristics prominence, should the operation be applied on cross_attention(I, T) or on I + cross_attention(I, T), where I is the image feature and T is the text embedding?

(4) I believe further experiments on widely used multi-instance generation benchmarks are necessary to better demonstrate the effectiveness of your method.

[1] Attend-and-Excite: Attention-Based Semantic Guidance for Text-to-Image Diffusion Models.

[2] TRAINING-FREE STRUCTURED DIFFUSION GUIDANCE FOR COMPOSITIONAL TEXT-TO-IMAGE SYNTHESIS.

---

> ### Author Response · Authors · 2024-11-20
> **Response to Reviewer wtqN**
>
> Dear Reviewer wtqN,
>
> Thank you for your constructive and insightful comments for helping us improve our paper. Please see below our response to your specific questions.
>
> ***
> `Q1: Transfer prompt balance to other multi-instance T2I scenarios, such as AAE and StructureDiff.`
>
> A1: Thank you for the suggestion. Following your suggestion, we have added experimental results on transferring the Prompt Balance module to Attend-and-Excite. The results are shown in **Figure 10 of Appendix F**.  Despite the limitations of SD V1.4, the PB module effectively balances the embedding strength between instances without layout guidance, enhancing their representation. While both instances are successfully generated in most cases, some attributes become entangled, resulting in artifacts like a cat with frog legs or a pig with bird wings. This nonetheless demonstrates the PB module's effectiveness, given the lack of spatial separation for the instances. We will explore other experiments when time is allowed.
>
> ***
> `Q2: Explanation for the CP module and the same type of multiple instances with different attributes.`
>
> A2: (1) There may be a misunderstanding here. The setting of $K$ is not strictly tied to the number of instances. In a feature space with an embedding of 1024, corresponding to 1024 feature maps, the top 2 strongest patterns in all maps can encompass all instances, especially after Prompt Balance, which significantly increases the likelihood of instance selection. This is discussed in our Top $K$ analysis in **Appendix B**, where we provide statistics on the probability of each instance being selected. **Lines 521-527** also mention that increasing $K$ initially improves generation quality but eventually introduces more noise, as shown in **Figure 8**.
>
> (2) Generating instances of the same type with different attributes is indeed challenging for multi-instance synthesis. To verify this, we have designed three prompts, including one group of five colored roses and two groups of five colored cats. As shown in **Figure 14 (c)**, our approach accurately assigns distinct properties to identical instances. However, for uncommon attributes like a red cat, the model struggles due to limitations in the original SDXL, though it successfully generates a green cat as prompted.
>
>
> ***
> `Q3: Details of characteristics prominence, whether "applied on cross_attention(I, T)" or "on I + cross_attention(I, T)".`
>
> A3: Thank you for the careful reading. Our characteristics prominence module is applied on cross attention(I, T). To avoid confusion, we have added the explanation in Line 335 to make it clearer.
>
> ***
> `Q4: Quantitative experiments and user study.`
>
> A4: Thank you for the suggestion. Given that our current approach involves sketch-based multi-instance scene generation, existing benchmarks should be adjusted for our evaluation, such as adding sketch inputs for T2ICompBench (Huang et al., 2023). Therefore, we have designed 20 sketches of complex scene composition, each with more than four sub-prompts, encompassing various terrains (plains, mountains, deserts, glaciers, cities) and diverse instances (rivers, bridges, stones, castles). We have utilized CLIP-Score (Hessel et al., 2021) for the global prompt and image, and evaluated the CLIP-Score for each background prompt and instance prompts by cropping the corresponding regions. Additionally, we have conducted a user study to assess different variants of our approach, using a 1-5 rating scale to evaluate image quality, placement, and prompt-image consistency. Details can be found in **"Metrics" of Section 5.1, "Quantitative Evaluation" of Section 5.2, and Appendix E**. The results of quantitative experiments and user studies are shown below, both of which demonstrate good improvements of our T$^3$-S2S approach over the ControlNet model and variants:
>
> | Model       | Global $\uparrow$   | Instances $\uparrow$ | Background $\uparrow$ | User $\uparrow$ |
> | ----------- | ---------- | ----------- | ------------ | ------ |
> | ControlNet  | 0.3440     | 0.2423      | 0.2539       | 2.34   |
> | PB          | 0.3447     | 0.2479      | 0.2562       | 2.64   |
> | DT          | 0.3433     | 0.2463      | 0.2549       | 3.29   |
> | CP          | 0.3467     | 0.2490      | 0.2563       | 3.45   |
> | PB+CP       | 0.3465     | 0.2548      | 0.2573       | 3.61   |
> | T$^3$-S2S   | **0.3497** | **0.2563**  | **0.2588**   | **3.88** |
> **Table**: Comparison of CLIP-Score across several variants, evaluated on whole images, masked instance regions, and masked background regions. Includes user study ratings on a scale of 1-5.
>
> ***
>
> Best regards,
>
> Authors

---

> ### Author Response · Authors · 2024-11-25
> **Follow-Up on Rebuttal and Reviewer Feedback**
>
> Dear Reviewer,
>
> I hope this message finds you well.
>
> First of all, I would like to express my heartfelt gratitude for the detailed and insightful comments you provided during the review process. From your feedback, we can clearly see that you thoroughly read our manuscript and offered highly constructive suggestions, which have been immensely valuable in improving our work. Following your comments, we have conducted additional experiments and made significant revisions to our draft, which we believe have greatly strengthened the manuscript.
>
> We submitted our rebuttal and the revised manuscript a few days ago and are eagerly awaiting your thoughts. Your feedback is crucial not only for ensuring that all concerns are addressed but also for presenting our work in the best possible light.
>
> It is especially encouraging to know that you are currently one of the reviewers who has given us a positive evaluation, and your perspective gives us hope. We deeply appreciate your constructive input and would be incredibly grateful if you could continue to support us by reviewing the revisions and providing further encouragement. Please trust in your judgment, as it is vital for us during this critical stage.
>
> Thank you again for your time and effort in guiding us through this process. We look forward to hearing your thoughts soon.
>
> Best regards,
>
> Authors

---

> ### Comment · Reviewer_wtqN · 2024-11-26
>
> Thanks to the authors for their serious reply, which solved my concern.
>
> **I decided to raise my score to 8.**
>
> In addition, I hope the authors can open-source the code in the future, which will be beneficial to the development of the community.

---

> ### Author Response · Authors · 2024-11-26
>
> Dear Reviewer wtqN,
>
> Thank you for your thoughtful response and for addressing my concern.
>
> This is the best news we’ve received recently, and we truly appreciate your decision to increase the score to 8.
>
> We are committed to open-sourcing our code in the future, as we believe it will greatly benefit the development of the community. Your encouragement further motivates us to make this happen.
>
> Thank you once again for your support and recognition!
>
> Best regards,
>
> Authors

---

### Official Review · Reviewer_7JUo · 2024-10-31

**Soundness:** 3
**Presentation:** 2
**Contribution:** 2
**Rating:** 5
**Confidence:** 4

**Summary:**

This paper identifies a phenomenon in sketch-to-scene generation where certain instance are not effectively represented in complex, multi-instance scenes. The authors attribute this issue to the imbalance of prompt energy and the homogeneity of value matrices. To address these challenges, the paper proposes a method that incorporates prompt balance, characteristic prominence, and dense tuning for sketch-to-image generation in complex scenes.

**Strengths:**

1. The paper provides a thorough analysis of text embeddings and the value matrices in cross-attention, revealing the issues of prompt energy imbalance and value matrix homogeneity.

2. The proposed method seemly  addresses the challenges associated with sketch-to-scene generation involving multiple subjects.

**Weaknesses:**

My primary concern lies with the clarity of the writing and the evaluation methodology:

1. The writing lacks clarity and the notation system is somewhat confusing.
2. Although the method does not require training, the optimization during inference may increase time demands, yet the paper does not report specific inference times.
3. The results are solely qualitative, lacking quantitative comparisons. The effectiveness of the proposed modules is not validated with measurable metrics, raising concerns about potential cherry-picking in qualitative assessments. While it may be challenging to establish reasonable quantitative benchmarks for this task, there even is no mention of conducting user studies to support the findings.

**Questions:**

see weakness.

---

> ### Author Response · Authors · 2024-11-20
> **Response to Reviewer 7JUo**
>
> Dear Reviewer 7JUo,
>
> Thank you for your constructive and insightful comments for helping us improve our paper. Please see below our response to your specific questions.
>
> ***
> `Q1: The clarity of the writing and notation`
>
> A1: Thank you for the helpful comment. We have improved the clarity, presentation and notation to make our contributions more clearly.
>
> ***
> `Q2: Comparison of specific Inference time.`
>
> A2: Thank you for your constructive comment. All our operations are performed using parallel matrix calculations, and our approach remains unaffected by the number of instances, ensuring stable speed. The inference time for a single prompt increases slightly from 18s to 29s on a single Nvidia V100 GPU. There is still room for optimization, such as converting matrix operations to lower-level CUDA operations, which could improve speed. We plan to explore this optimization in the future.
>
> ***
> `Q3: Quantitative experiments and user study.`
>
> A3: We agree with you that it may be challenging to establish reasonable quantitative benchmarks for this task. So instead of conducting experiment on the existing benchmark, we have designed 20 sketches of complex scene composition, each with more than four sub-prompts, encompassing various terrains (plains, mountains, deserts, glaciers, cities) and diverse instances (rivers, bridges, stones, castles). We have utilized CLIP-Score (Hessel et al., 2021) for the global prompt and image, and evaluated the CLIP-Score for each background prompt and instance prompts by cropping the corresponding regions. Additionally, we have conducted a user study to assess different variants of our approach, using a 1-5 rating scale to evaluate image quality, placement, and prompt-image consistency. Details can be found in **"Metrics" of Section 5.1, "Quantitative Evaluation" of Section 5.2, and Appendix E**. The results of quantitative experiments and user studies are shown below, both of which demonstrate good improvements of our T$^3$-S2S approach over the ControlNet model and variants:
>
>
> | Model       | Global $\uparrow$   | Instances $\uparrow$ | Background $\uparrow$ | User $\uparrow$ |
> | ----------- | ---------- | ----------- | ------------ | ------ |
> | ControlNet  | 0.3440     | 0.2423      | 0.2539       | 2.34   |
> | PB          | 0.3447     | 0.2479      | 0.2562       | 2.64   |
> | DT          | 0.3433     | 0.2463      | 0.2549       | 3.29   |
> | CP          | 0.3467     | 0.2490      | 0.2563       | 3.45   |
> | PB+CP       | 0.3465     | 0.2548      | 0.2573       | 3.61   |
> | T$^3$-S2S   | **0.3497** | **0.2563**  | **0.2588**   | **3.88** |
> **Table**: Comparison of CLIP-Score across several variants, evaluated on whole images, masked instance regions, and masked background regions. Includes user study ratings on a scale of 1-5.
>
> ***
>
> Best regards,
>
> Authors

---

> > ### Comment · Reviewer_7JUo · 2024-11-28
> >
> > Thank you for the author's response, most of my concerns have been addressed. However, I agree with other reviewers' concerns about the novelty of the methods presented in this paper. For me, this paper mainly focuses on engineering tuning rather than presenting significant novel techniques. Additionally, using only the CLIPScore metric may be too thin; it is unclear whether the proposed method would harm image quality or aesthetic scores. Therefore, I will maintain my current rating.

---

> ### Author Response · Authors · 2024-11-25
> **Follow-Up on Rebuttal and Reviewer Feedback**
>
> Dear Reviewer,
>
> I hope this message finds you well.
>
> First of all, we sincerely thank you for your detailed and insightful comments during the review process. Your feedback has been incredibly valuable in guiding us to strengthen our work.
>
> In particular, your observation about the challenge of establishing reasonable quantitative benchmarks for this task was especially insightful. To address this, we designed quantitative experiments based on 20 complex cases and conducted a user study. These additions have enhanced our paper and allowed us to better evaluate the proposed method.
>
> We submitted our rebuttal and the revised manuscript a few days ago and are eager to hear your thoughts. Your feedback is essential to ensure that all concerns have been fully addressed. While we understand your time is limited, we would greatly appreciate it if you could review our responses and let us know if further clarifications are needed.
>
> Thank you again for your time and effort throughout this process. Your positive evaluation would mean a great deal to us, and we are committed to continuously improving our work based on your valuable guidance.
>
> Best regards,
>
> Authors.

---

> ### Author Response · Authors · 2024-11-28
> **Follow-Up Response to Reviewer 7JUo**
>
> Dear Reviewer 7JUo,
>
> Thank you for your response.  Below, we will address the two remaining points that you raised.  While previous studies have primarily focused on the influence of attention maps on generation, our work aims to explore  the roles of text embeddings and the Value matrix within the attention mechanism. Through a thorough analysis and tests, we developed Training-free Triplet Tuning, a method that addresses the entire cross-attention process from a holistic perspective. This effectively tackles the challenges associated with generation tasks involving multiple subjects, as you also echoed in your assessment of our work’s strengths.
>
> CLIPScore is a widely accepted metric for assessing text-image consistency.   In addition to that, we also conducted user studies to give subjective assessments, including aesthetic quality and text-image consistency as you suggested previously.   These two aspects assess both the visual appeal and the alignment with user expectations, offering a complementary perspective to the quantitative results. Additionally, numerous visualization cases in Appendix show high-quality image generation across diverse scenes, including black-and-white sketches, game-style scenes, and realistic scenes, demonstrating the stability of our method.
>
> We hope this clarification better addresses your concerns regarding the novelty and evaluation of our method. We sincerely appreciate your constructive feedback and your role in helping us improve the presentation of this paper.
>
> Best regards,
>
> Authors

---

### Official Review · Reviewer_JFtx · 2024-11-01

**Soundness:** 2
**Presentation:** 2
**Contribution:** 2
**Rating:** 5
**Confidence:** 5

**Summary:**

This article primarily focuses on the task of sketch-to-scene generation. Specifically, for complex scenes with multiple detailed objects, previous methods sometimes miss small or uncommon instances. Addressing this issue, the article proposes a training-free triplet adjustment method for sketch-to-scene generation. This method can be directly applied to existing SDXL and ControlNet models, enabling them to effectively tackle the multi-instance generation problem, including prompt balancing, feature highlighting, and dense adjustment. The proposal is made after reviewing the entire cross-attention mechanism. This solution revitalizes the existing ControlNet model, allowing it to effectively handle multi-instance generation, involving prompt balancing, feature highlighting, and dense adjustment. Experiments demonstrate that this triplet adjustment method enhances the performance of existing sketch-to-image models, enabling the generation of detailed, multi-instance 2D images that closely follow input prompts and enhance visual quality in complex multi-instance scenes.

**Strengths:**

1. This article addresses a very meaningful area, utilizing the approach of scene generation through sketches.

2. The method requires no training and can be seamlessly integrated into existing controllable generation models, thereby enhancing the models' generative performance.

**Weaknesses:**

1. The experimental content of this article is somewhat lacking in depth. From the presented experimental results, it only includes the generation of isometric view of game scene using colored sketch lines as control inputs. This raises the question of whether the method proposed in this article has universal applicability to scene sketch generation tasks.

2. According to the description in this article, this training-free method at the feature level should be universally applicable to various controllable generation models. The method proposed in this article is based on ControlNet, but the article does not sufficiently discuss whether this approach would still be effective when integrated into other controllable generation models.

**Questions:**

1. Since this method can enhance the performance of existing controllable generation models without the need for training, is it possible to include experimental analyses based on more diverse existing models, such as the T2I-Adapter (https://arxiv.org/abs/2302.08453)   that the article compares?
If additional experiments could be added to demonstrate this, it would better showcase the universal applicability of the method proposed in the article, significantly enhancing its contribution.

2. Can the method proposed in this article be applied to a wider range of scenarios? From the experimental results presented in the article, it seems that simply altering the prompt should be sufficient to produce a more diverse array of scene images. I hope to see more experimental outcomes. Is it possible to incorporate a wider variety of input types (for example, black and white hand-drawn sketches)? Can the output be expanded to include a richer variety of scene images (for example, real-world images), not just limited to the isometric view of game scenes?
Particularly, the experimental results in Figure 6 of the article suggest that the method's control over the style of generated images should stem solely from the specific input prompt "Isometric view of game scene." Therefore, if the article could showcase more use cases across a variety of application scenarios, it would greatly increase the value of the method proposed in the article.

---

> ### Author Response · Authors · 2024-11-20
> **Response to Reviewer JFtx**
>
> Dear Reviewer JFtx,
>
> Thank you for your constructive and insightful comments for helping us improve our paper. Please see below our response to your specific questions.
>
> ***
> `Q1: Explanation for game scene phrases and colored sketches. `
>
>
> A1: Thank you for the suggestion. To ensure consistency, we have started each prompt with “Isometric view of a game scene” when generating controlled synthetic images for game settings. This helps maintain a consistent angle and style, ignoring any incoherent instance sketches that might appear in real-world scenes, thereby focusing on object placement and verifying text-image consistency. The colored sketches are used solely to distinguish between different instances, and the colors used are arbitrary without class or semantic information. To validate this, we have also used grayscale sketches as input, and the resulting images are nearly identical under the same random seed (two columns pointed by the red arrows in **Figures 12, 13 and 14**).
>
> ***
> `Q2: Diversity of generated scenes, including removing game scene phrases, real-world scenarios, and black-and-white sketches. `
>
>
> A2: To verify that our approach is not limited to game scenes, (1) We have included more results on sketches of complex composition in **Figures 12 and 13 of Appendix C**, each with more than four sub-prompts, encompassing various terrains (plains, mountains, deserts, tundra, cities) and diverse instances (rivers, bridges, stones, castles); (2) We have added results of diverse scenes in **Figure 14 of Appendix D** to verify multi-scenario applicability, including a) four common simple scenes; b) two indoor scenes; and c) three scenes featuring instances of the same type but with different color attributes. Among these diverse scenes, our approach demonstrates superior performance with improved fidelity and precision in aligning with text prompts and sketch layouts for multi-instance synthesis.
>
> ***
> `Q3: Universally applicable to various controllable generation models, such as the T2I-Adapter. `
>
> A3: Thank you for the suggestion. To validate the general applicability of our approach beyond the ControlNet model, we have applied T$^3$-S2S to the T2I-Adapter model. We have used grayscale sketches for quick validation, which contain less semantic information. The PB and CP modules remain unchanged, while the DT module is integrated into the SDXL main channel, similar to CP, as it could not be placed in a separate branch like in ControlNet. We have used the same prompts and sketches from the main results (Figure 6) and Appendix C, with all other hyperparameters unchanged. The results are presented in **Figure 11 of Appendix G**. T$^3$-S2S effectively improves T2I-Adapter's alignment with prompts and layouts in complex scenes, demonstrating promising generalizability and effective control across both ControlNet and T2I-Adapter models.
>
> ***
>
> Best regards,
>
> Authors

---

> ### Author Response · Authors · 2024-11-25
> **Follow-Up on Rebuttal and Reviewer Feedback**
>
> Dear Reviewer,
>
> I hope this message finds you well.
>
> First of all, we sincerely thank you for your detailed and insightful comments during the review process. Your valuable suggestions have greatly guided us, and we have worked diligently to address all concerns.
>
> In particular, your suggestions regarding validating the T2I adapter and exploring diversity in black-and-white sketch scenarios have been instrumental in demonstrating the generalizability of our proposed method. We conducted several additional experiments as requested, which we believe have significantly strengthened our manuscript.
>
> We submitted our rebuttal and the revised manuscript a few days ago and are eager to hear your thoughts. Your feedback is crucial to us, and we want to ensure that all concerns are fully addressed. We understand your time is limited, but we would deeply appreciate it if you could review our responses and let us know if further clarifications are needed.
>
> Thank you again for your time and effort throughout this process. Your positive evaluation would mean a great deal to us, and we are committed to further improving our work based on your guidance.
>
> Best regards,
>
> Authors.

---

### Official Review · Reviewer_s5sU · 2024-11-04

**Soundness:** 3
**Presentation:** 3
**Contribution:** 2
**Rating:** 6
**Confidence:** 4

**Summary:**

The paper proposes a Training-free Triplet Tuning (T³-S2S) method for sketch-to-scene generation, aiming to enhance the quality of generated scenes from sketches without additional training. The authors identify challenges in existing diffusion models related to imbalanced prompt energy and value homogeneity in the cross-attention mechanism, which lead to missing or coupled instances in complex scenes. To address these issues, they introduce two strategies: Prompt Balance and Characteristics Prominence. Additionally, they incorporate Dense Tuning from Dense Diffusion to refine attention maps. The proposed method is evaluated qualitatively on game scene generation tasks, demonstrating improvements in generating detailed, multi-instance scenes that align with input sketches and prompts.

**Strengths:**

1. The proposed Prompt Balance and Characteristics Prominence strategies contribute to enhancing the quality of generated scenes, particularly in handling complex scenes with multiple instances.
2. The method is relatively simple and does not require additional training, making it efficient and practical.

**Weaknesses:**

1. The proposed Prompt Balance and Characteristics Prominence strategies appear to be incremental improvements on existing techniques such as TopK methods. Dense Tuning is adapted from Dense Diffusion. Therefore, the paper may lean more towards engineering optimization rather than presenting significant novel techniques.
2. The experimental comparisons use models based on different versions of Stable Diffusion, with the proposed method and T2I Adapter using SDXL (a more advanced model), while Dense Diffusion is based on SD v1.5. Since SDXL offers higher generation quality than SD1.5, this results in an unfair comparison. Considering that this work draws inspiration from Dense Diffusion, it would be more appropriate to either adapt Dense Diffusion to SDXL or apply the proposed method to SD1.5 to ensure a fair evaluation.
3. The experiments focus primarily on game scenes, with limited variety in the types of scenes presented. Some examples are repeated in the paper, and the lack of diverse examples may not fully demonstrate the method's generalizability to other contexts.
4. The paper focuses heavily on qualitative analysis and lacks quantitative experiments.
5. Line 360 and 361: citations of SDXL and ControlNet are incorrect.

**Questions:**

1. I wonder whether the proposed method is effective for other types of scenes. Exploring and presenting results on different scene categories would be beneficial.
2. I suggest to incorporate user studies to enhance the robustness of the experimental validation.
3. The appendix seems to be directly attached after the references. It might be more appropriate to format the appendix according to the conference guidelines, ensuring it is properly integrated or submitted as a supplementary document as required.

---

> ### Author Response · Authors · 2024-11-20
> **Response to Reviewer s5sU -- Part1**
>
> Dear Reviewer s5sU,
>
> Thank you for your constructive and insightful comments for helping us improve our paper. Please see below our response to your specific questions.
>
> ***
> `Q1: Correlation between PB and CP with the TOP $K$ method, and Explanation for Dense Tuning.`
>
> A1: TOP-K is a versatile concept used across domains to select the most relevant elements, enhancing efficiency and focus. In transformer architectures [1-4], it improves efficiency by selecting the top K attention scores, creating sparse attention, and ignoring the weak maps. In our paper, our method uses the concept of top-k not just for selection but to balance and highlight each instance and its attributes in the feature map. Prompt Balance adjusts the energy distribution of instance-related embeddings in the global prompt, while Characteristics Prominence uses Top-K selection to emphasize instance-specific attributes, enhancing the relevant feature map channels (**Lines 76-89**). We apply dense tuning (Kim et al., 2023) in the ControlNet branch to refine attention map contours, strengthen instance regions, and enhance the modulation of the entire cross-attention process. Experimental evaluations demonstrate that our T3-S2S approach improves text-to-image models, consistently producing detailed, multi-instance scenes aligned with input sketches and prompts.
>
>
> [1] Memory-efficient Transformers via Top-k Attention.
>
> [2] Statistical Perspective of Top-K Sparse Softmax Gating Mixture of Experts.
>
> [3] Sparser is Faster and Less is More: Efficient Sparse Attention for Long-Range Transformers.
>
> [4] Explicit Sparse Transformer: Concentrated Attention Through Explicit Selection.
>
>
> ***
> `Q2: Fair comparison for Dense Diffusion.`
>
> A2: Thank you for the suggestion. We have applied the Dense Diffusion to SDXL model for a fair comparison in **the "Baselines" of Section 5.1 and Figure 6**, which have better generation quality than the base on the SD V1.5.
>
> ***
> `Q3: Diversity of generated scenes, presenting results on different scene categories.`
>
> A3: To verify that our approach can generate diverse scenes, (1) We have included more results on sketches of complex scene composition in **Figures 12 and 13 of Appendix C**, each with more than four sub-prompts, encompassing various terrains (plains, mountains, deserts, tundra, cities) and diverse instances (rivers, bridges, stones, castles); (2) We have added results of diverse scenes in **Figure 14 of Appendix D** to verify multi-scenario applicability, including a) four common simple scenes; b) two indoor scenes; and c) three scenes featuring instances of the same type but with different color attributes. Among these diverse scenes, our approach demonstrates superior performance with improved fidelity and precision in aligning with text prompts and sketch layouts for multi-instance synthesis.
>
> ***
> `Q4: Quantitative experiments and user study.`
>
> A4: Thank you for the suggestion. We have designed 20 sketches of complex scene composition, each with more than four sub-prompts, encompassing various terrains (plains, mountains, deserts, glaciers, cities) and diverse instances (rivers, bridges, stones, castles). We have utilized CLIP-Score (Hessel et al., 2021) for the global prompt and image, and evaluated the CLIP-Score for each background prompt and instance prompts by cropping the corresponding regions. Additionally, we have conducted a user study to assess different variants of our approach, using a 1-5 rating scale to evaluate image quality, placement, and prompt-image consistency. Details can be found in **"Metrics" of Section 5.1, "Quantitative Evaluation" of Section 5.2, and Appendix E**. The results of quantitative experiments and user studies are shown below, both of which demonstrate good improvements of our T$^3$-S2S approach over the ControlNet model and variants:
>
> ***
> | Model       | Global $\uparrow$   | Instances $\uparrow$ | Background $\uparrow$ | User $\uparrow$ |
> | ----------- | ---------- | ----------- | ------------ | ------ |
> | ControlNet  | 0.3440     | 0.2423      | 0.2539       | 2.34   |
> | PB          | 0.3447     | 0.2479      | 0.2562       | 2.64   |
> | DT          | 0.3433     | 0.2463      | 0.2549       | 3.29   |
> | CP          | 0.3467     | 0.2490      | 0.2563       | 3.45   |
> | PB+CP       | 0.3465     | 0.2548      | 0.2573       | 3.61   |
> | T$^3$-S2S   | **0.3497** | **0.2563**  | **0.2588**   | **3.88** |
> **Table**: Comparison of CLIP-Score across several variants, evaluated on whole images, masked instance regions, and masked background regions. Includes user study ratings on a scale of 1-5.

---

> ### Author Response · Authors · 2024-11-20
> **Response to Reviewer s5sU -- Part2**
>
> ***
> `Q5: Some typos, such as citations.`
>
> A5: Thank you for your careful reading. We have corrected them in Lines 358 and 359.
>
> ***
> `Q6: The position of the appendix.`
>
> A6: Thank you for the suggestion. As ICLR 2025 guidelines and FAQ state, 'We encourage authors to submit a single file (paper + supplementary text)' and 'you can include the appendices at the end of the main pdf after the references'. To address potential issues with the overly large PDF, we have compressed images without compromising clarity for smoother loading. If you feel that it is better to have a separate file for the appendix, we will split it into standalone supplementary material later.
>
> ***
>
> Best regards,
>
> Authors

---

> ### Author Response · Authors · 2024-11-25
> **Follow-Up on Rebuttal and Reviewer Feedback**
>
> Dear Reviewer,
>
> I hope this message finds you well.
>
> First of all, we sincerely thank you for your detailed and insightful comments during the review process. Your comprehensive feedback has been incredibly valuable for improving our paper. In particular, your suggestions regarding the quantitative experiments and Dense fair comparison have significantly helped us refine multiple aspects of our work.
>
> We have worked diligently to address all your comments and conducted additional experiments as requested, which we believe have further strengthened our manuscript. We submitted our rebuttal and the revised version a few days ago and are eager to hear your thoughts.
>
> Your feedback is essential to us, and we want to ensure that all concerns are fully addressed. While we understand your time is limited, we would greatly appreciate it if you could review our responses and let us know if further clarifications are needed.
>
> Thank you once again for your time and effort throughout this process. Your positive evaluation would mean a great deal to us, and we remain committed to continuously improving our work based on your guidance.
>
> Best regards,
>
> Authors.

---

> ### Author Response · Authors · 2024-12-03
> **Thanks you for support**
>
> Dear Reviewer s5sU,
>
> Thank you very much for recognizing and acknowledging our work.
>
> We truly appreciate your positive feedback and for raising the score to 6—it means a lot to us.
>
> Best regards,
>
> Authors.

---

### Author Response · Authors · 2024-11-20
**General response**

Dear reviewers and meta reviewers,

Thank you for your valuable feedback (we refer Reviewer s5sU as **R1**, JFtx as **R2**, 7JUo as **R3**, wtqN as **R4**).
We are glad to see your positive comments on our method, e.g., "Contribute to the quality" (**R1**, **R2**), "Handling complex scenes" (**R1**), "Relatively simple" (**R1**), "Training-free" (**R1**, **R2**), "Efficient" (**R1**), "Practical"(**R1**), "Addresses a very meaningful area" (**R2**), "Seamlessly integrated"(**R2**), "Provide a thorough analysis" (**R3**), "Addresses the challenges of S2S" (**R3**), "Certain innovations" (**R4**) and "Indeed yields notable improvements" (**R4**).

According to your valuable comments and suggestions, we have thoroughly revised our manuscript by adding more details and studies, as part of the ICLR revision process. In particular, we created and tested 20 complex scene sketches, each with over four sub-prompts, covering various terrains (plains, mountains, deserts, glaciers, cities) and diverse elements (rivers, bridges, stones, castles). We conducted quantitative evaluations tailored to sketch-to-scene scenarios using both CLIP-Score and user studies. Additionally, we included results for diverse scenes to validate multi-scenario applicability, including: a) common simple scenes; b) indoor scenes; and c) scenes featuring instances of the same type but with different color attributes. Through these quantitative results and visualizations, our approach demonstrates superior performance, achieving higher fidelity and precision in aligning text prompts and sketch layouts for multi-instance synthesis.

We understand that this may add to your workload, but we sincerely appreciate your time and efforts in reviewing our improvements based on your feedback. Please also find our specific response to each reviewer below.

Best regards,

Authors

***
`Additionally, the changes have been highlighted using the blue font in the revised paper, and we will release our code in the camera-ready version. The details of the revision are summarized as follows:`

* We have implemented quantitative experiments and user studies using commonly used metrics in **"Metrics" of Section 5.1, "Quantitative Evaluation" of Section 5.2, and Appendix E**, both of which demonstrate good improvements of our T$^3$-S2S approach over the ControlNet model (**R1/R3/R4**).

* We have added more results for different sketches of complex scene composition in **Figures 12 and 13 of Appendix C**, each with more than four sub-prompts, encompassing various terrains (plains, mountains, deserts, tundra, cities) and diverse instances (rivers, bridges, stones, castles) (**R1/R2**).

* We have added more results of diverse scenes in **Figure 14 of Appendix D** to verify multi-scenario applicability, including (1) four common simple scenes; (2) two indoor scenes; and (3) three scenes featuring instances of the same type but with different color attributes (**R1/R2/R4**).

* We have experimented with transferring T$^3$-S2S to T2I-Adapter in **Figure 11 of Appendix G**, and have validated the general applicability and performance improvements using our approach beyond the ControlNet model (**R2**).

* We have experimented with transferring the Prompt Balance module to Attend-and-Excite in **Figure 10 of Appendix F**, to further validate the effectiveness of the PB module (**R4**).

* We have applied the SDXL model to Dense Diffusion for a fair comparison in **the "Baselines" of Section 5.1 and Figure 6** (**R1**).

* We have fixed several typos, such as citations and the tense (**R1/R3**).

* We have moved the Top K analysis from the main text to **Appendix B**.

---

> ### Author Response · Authors · 2024-11-21
> **Authors rebuttal posted.   Awaiting for reviewers'  responses.**
>
> Dear Reviewers, SAC and AC,
>
> We have carefully responded to reviewers' every major comment, a few days ago,  and uploaded a revised version following reviewers'  suggestions.  Please take a look at our detailed responses (both general, and individualized) and the revision,   to see if your questions have been addressed,  or ask further clarifying questions where needed before the due date, so that we can provide follow-up responses in a timely fashion (considering the time-lag due to time-zone difference).  Thank you !!
>
> Best regards,
> Authors.

---

### Comment · Area_Chair_zcs2 · 2024-11-28
**Reviewer feedback on rebuttal**

Dear Reviewers,

As the discussion period will end next week, please take some time to read the authors' rebuttal and provide feedback as soon as possible. For reviewers s5sU, JFtx, and 7JUo, did the author rebuttal address your concerns, and do you have further questions?

Thanks,
Area Chair

---

> ### Author Response · Authors · 2024-12-03
> **Thanks for Your Guidance**
>
> Dear Area Chair zcs2,
>
> With the deadline approaching, thank you very much for your efforts in facilitating the discussion and guiding the review process.
>
> We truly appreciate your support.
>
> Best regards,
>
> Authors

---

### Author Response · Authors · 2024-12-01
**Response and Appreciation**

Dear Reviewers, AC, and SAC,

Thank you all for your valuable comments on our paper. As the rebuttal process draws to a close, we are delighted that our responses have been able to address most of the reviewers' concerns, as no further questions have been raised. We would also like to sincerely thank the AC for actively facilitating discussions during this process.

We deeply appreciate your time and effort in guiding us through this journey.

Thank you once again for your support and feedback.

Best regards,

Authors

---

### Note · Authors · 2025-01-23

I have read and agree with the venue's withdrawal policy on behalf of myself and my co-authors.